# Two-Dimensional MoS_2_: Structural Properties, Synthesis Methods, and Regulation Strategies toward Oxygen Reduction

**DOI:** 10.3390/mi12030240

**Published:** 2021-02-27

**Authors:** Hanwen Xu, Jiawei Zhu, Qianli Ma, Jingjing Ma, Huawei Bai, Lei Chen, Shichun Mu

**Affiliations:** 1State Key Laboratory of Advanced Technology for Materials Synthesis and Processing, Wuhan University of Technology, Wuhan 430070, China; xuhw@whut.edu.cn (H.X.); zhujiawei@whut.edu.cn (J.Z.); maqianli@whut.edu.cn (Q.M.); majingjing@whut.edu.cn (J.M.); 2381728231@whut.edu.cn (H.B.); 2Foshan Xianhu Laboratory of the Advanced Energy Science and Technology, Guangdong Laboratory, Xianhu Hydrogen Valley, Foshan 528200, China

**Keywords:** electrolysis, oxygen reduction reaction, 2D materials, catalyst, MoS_2_

## Abstract

Compared with three-dimensional (3D) and other materials, two-dimensional (2D) materials with unique properties such as high specific surface area, structurally adjustable band structure, and electromagnetic properties have attracted wide attention. In recent years, great progress has been made for 2D MoS_2_ in the field of electrocatalysis, and its exposed unsaturated edges are considered to be active sites of electrocatalytic reactions. In this review, we focus on the latest progress of 2D MoS_2_ in the oxygen reduction reaction (ORR) that has not received much attention. First, the basic properties of 2D MoS_2_ and its advantages in the ORR are introduced. Then, the synthesis methods of 2D MoS_2_ are summarized, and specific strategies for optimizing the performance of 2D MoS_2_ in ORRs, and the challenges and opportunities faced are discussed. Finally, the future of the 2D MoS_2_-based ORR catalysts is explored.

## 1. Introduction

Due to the high energy conversion efficiency and environmental friendliness, hydrogen fuel cells have been regarded as one of the most important clean energy conversion devices to solve the energy crisis and environmental pollution. However, the main challenge in hydrogen fuel cells is to find a highly efficient catalyst for the oxygen reduction reaction (ORR) [1]. The complex multi-step electron transfer process and the slow mass diffusion of the reaction itself lead to slow ORR kinetics [2]. Hitherto, platinum (Pt)-based catalysts are still considered as the most efficient ORR catalysts, but the high cost and poor durability severely limit their wide applications [3]. Therefore, these disadvantages have triggered intense interest in non-Pt catalysts with both high ORR activity and stability. Currently, a variety of non-platinum metal-based ORR catalysts, such as defective carbon materials [4], heteroatom-doped metal-free carbon [5], transition metal/heteroatom-co-doped carbon [6,7], intermetallic compounds [8], and metal nitrides [9], have been synthesized. Among them, two-dimensional (2D) materials with atomic or molecular thickness and infinite planar length exhibit peculiar physicochemical properties, including high specific surface area, adjustable band structure, and electromagnetic properties, which make 2D materials become one of the most important non-Pt-based ORR catalysts [10,11,12,13].

A variety of 2D materials have been found for electrocatalysis. Among heteroatom-doped carbon [14], layered metal hydroxide (LDH) [15], layered metal oxide [16], 2D metal-organic frameworks (2D-MOFs) [17], 2D covalent organic frameworks (2D-COFs) [18], graphite carbonitrides (g-C3N4) [19], hexagonal boron nitride (h-BN) [20], black phosphorous (BP) [21], MXenes [22], and 2D transition metal dichalcogenides (TMDs) [23,24,25], TMDs are a class of materials with great application prospects and basic research value. The chemical formula of TMDs is MX2, where M represents transition metal elements, including group IV (Ti, Zr, or Hf), group V (V, Nb, or Ta), group VI (Mo, W), group VII (Tc, Re), or group X (Pd, Pt), and X refers to a chalcogen atom (S, Se). In a TMD, X-M-X units stack on each other to form a sandwich structure, and the layers are combined by van der Waals forces [26].

As the thickness decreases, 2D TMDs exhibit a series of special properties. During the conversion process of TMDs from bulk to single-layered 2D materials, significant bandgap transitions occur, resulting in excellent photoelectric performance. Besides, due to the high degree of structural controllability and anisotropy of 2D materials, 2D TMDs can be controlled to form various shapes and sizes, doped with different heteroatoms, and their surfaces can be modified with other materials to obtain **a** unique crystal structure [27]. More importantly, there are unsaturated coordination and dangling bonds at the edges of TMDs nanosheets. These special properties of TMDs nanosheets provide many inspirations for basic research in many fields including catalysis [28,29,30,31,32,33], transistor [34], energy storage [35,36,37,38,39] and sensor [40,41,42]. Among all 2D TMDs, MoS_2_ is one of the few with a natural layered structure, indicating that MoS_2_ can be stripped to obtain high-quality 2D MoS_2_ without complicated chemical synthesis [26]. Therefore, the cost of preparing 2D MoS_2_ is much lower than other 2D TMDs due to the simple synthesis conditions. More importantly, 2D MoS_2_ is a 2D semiconductor with a direct band-gap, which has the best electric performance among 2D TMDs [43]. These make 2D MoS_2_ get more attention among 2D TMDs. 

So far, although there have been many reviews about MoS_2_ electrocatalysis, applying 2D MoS_2_ in ORRs, especially a detailed summary of the entire process from synthesis to application, is a little concerning. Thus, this review focuses on describing the ORR application of 2D MoS_2_ and its composite materials. Our aim is to provide comprehensive and cutting-edge information to deeply understand the electrocatalytic ORR for MoS_2_-based catalysts and reveal the relationship between structure and performance. For different types of MoS_2_ materials, we will discuss their structural characteristics and internal structure–activity relationship. We believe that electrocatalysts based on MoS_2_ could help to solve the energy problems we face. Finally, based on the current research results, the difficulty and deficiency encountered in the research are proposed, and the outlook is also discussed.

## 2. Structural Properties of 2D MoS_2_

The properties of 2D MoS_2_ are highly dependent on their crystal phase, structure size and chemical composition. In the MoS_2_ structure, the coordination of S elements and the stacking state between layers can form different electronic characteristics. The most common crystal structures of MoS_2_ are 1T and 2H phases, where the number represents the number of layers of the S-Mo-S layer in a unit cell, and T and H represent the tetragonal crystal system, and the hexagonal crystal system, respectively [44]. As shown in Figure 1A, in the 2H phase, each Mo atom exists in the center of the hexagonal prism coordination structure and covalently bonded to six S atoms, and exhibits semiconductor properties. In the 1T phase, one Mo atom and six S atoms form a twisted octahedral coordination structure, with metallicity [45]. In MoS_2_, the S-Mo-S of each planar unit is regarded as a single layer, which is composed of two layers of S atoms and an intermediate layer of Mo atoms [46]. Due to the weak van der Waals force connection between the layers, the peeling between the layers is easy to occur, which makes it possible to synthesize single-layer MoS_2_ [47].

As shown in Figure 1B, the bulk material of MoS_2_ is an indirect-bandgap semiconductor, in which the top of the valence band is located at point G and the bottom of the conduction band is located at the midpoint of the axis of symmetry of G–K, while in single-layer MoS_2_, the top of the valence band and the bottom of the conduction band coincide with point K, indicating that single-layer MoS_2_ is a direct-bandgap semiconductor [48]. This indirect-to-direct transition of the bandgap from bulk to the single-layered 2D material is caused by the quantum confinement effect. The electronic properties of 2D MoS_2_ are closely related to the d electrons existing in the d orbital between the bonding band and the anti-bonding band of the Mo–S bond. In 2H-MoS_2_, the non-bonded d orbitals are fully occupied, resulting in semiconductor properties. In 1T-MoS_2_, the non-bonded d orbitals are partially filled, resulting in higher conductivity. As the d-orbital electron occupancy rate decreases, 2D MoS_2_ gradually changes from a semiconductor to a conductor, which endows the material with different physical and chemical properties [27]. When the thickness of MoS_2_ drops to the atomic level, a large number of exposed surfaces can undergo oxidation or reduction reactions with various reagents [49]. These surface atoms can escape in the reaction to form vacancy defects, simultaneously leading to the disorder of the nanostructure and a reduction in the coordination number of surface atoms. By adjusting the vacancy defects, we can further modify the electronic characteristics of MoS_2_. At the same time, unsaturated edges and dangling bonds often appear on the edges of 2D MoS_2_. These unique edge defects also provide possible active sites for catalytic reactions [2,50]. 

So far, many 2D MoS_2_-based materials have been applied to ORRs. At present, it is generally believed that the active site of 2D MoS_2_ in the ORR is the unsaturated Mo site exposed at the edge. Because the Mo site is positively charged due to the possible polarization effect of the surrounding negatively charged S atoms, it can, therefore, be easily combined with negatively charged O atoms [51]. Simultaneously, the vacancy defects on the basal plane also contribute to ORR performance. In general, an excellent ORR electrocatalyst must have favorable stability, small overpotentials, high current densities, and low manufacturing cost [52]. Although MoS_2_ has a low cost and high chemical stability, its large inert basal plane and poor electron transport properties limit its catalytic activity. However, it is undeniable that after adjusting the structure of 2D MoS_2_, it still possesses unique advantages and broad development prospects for ORRs. Here are some advantages of 2D MoS_2_ in ORRs:

(1) The uncoordinated metallic edge centers not only provide a certain intrinsic ORR activity but also give an opportunity for effective functionalization with different groups.

(2) Compared with bulk MoS_2_, nanostructured MoS_2_ has a smaller particle size and a larger specific surface area. The small size 2D MoS_2_ is more sensitive to the catalytic reactions compared with bulk MoS_2_. At the same time, the large surface area can anchor more active sites [53]. Besides, 2D MoS_2_ has a unique open structure, which makes ORR reactants easily access the active site.

(3) 2D MoS_2_ with a uniformly exposed lattice plane can supply a much simpler condition as an ideal platform for combing experiments and theoretical results, which is helpful to further understand the origin of ORR activity.

## 3. The Synthesis Methods

### 3.1. Liquid-Phase-Stripping Method

Liquid-phase stripping is a common method for preparing monolayer and multilayer materials, usually based on ultrasound assistance (in a solvent or aqueous solution) and subsequent centrifugal process [43,54]. As shown in Figure 2, this method is extremely compatible. Since the MoS_2_ layers are connected by weak van der Waals forces, the 2D MoS_2_ layer can be peeled from bulk by weakening the force between the layers. Recently, an ion intercalation method was created for layered materials that can attract guest molecules between layers and form clathrates. First, intercalation agents such as n-butyllithium [55,56] and iodine bromide are used to increase the interlayer spacing, weaken the interlayer force, and reduce the energy barrier for peeling. Then, the layered material is peeled off by ultrasonic waves [55], thermal shock [57], and other treatments. Ambrosi et al. obtained complete MoS_2_ nanosheets under mild conditions by using a variety of Li-based intercalants and explored the effects of different intercalants on the material structure and catalysis performance. Among such Li-based intercalants (tert-butyllithium (t-Bu-Li), n-butyllithium (n-Bu-Li), and methyllithium (Me-Li)), the stripping efficiency of n-Bu-Li and t-Bu-Li is the highest, and the obtained MoS_2_ nanosheets have higher catalytic performance [56]. However, this liquid-phase intercalation method is susceptible to changes in the reaction environment, causing structural deformation of some materials [58,59]. Another problem is the re-aggregation of the material during the removal of the intercalant. 

To overcome the shortcomings of the intercalation method, such as time-consuming process and product instability, the latest stripping strategy is to place MoS_2_ in a solvent with a specific surface energy and then use ultrasonic waves to destroy the material crystallites to produce nanosheets. It was discovered that organic solvents such as N-cyclohexyl-2-pyrrolidone (CHP) [60] and N-methyl-2-pyrrolidone (NMP) [43,61] can effectively strip MoS_2_. With further exploration, it was found that using surfactants such as sodium cholate [62,63] as a stabilizer can have a synergistic effect with the solvent and effectively prevent the nanosheets from re-aggregating (Figure 3A). For the first time, Peter et al. employed a series of polymers (Polybutadiene-styrene (PBS), Polystyrene (PS), polyvinyl alcohol (PVA), Polyvinyl chloride (PVC), etc.) as stabilizers to replace surfactants, which proved that MoS_2_ could be stripped and could remain stable in a mixed solution of a variety of organic solvents and polymers (Figure 3B,C). More importantly, by a proposed simple model, they predicted that polymer stabilization is effective only when the solubility parameters of the nanosheets, polymer, and solvent are similar [64].

### 3.2. Hydrothermal/Solvothermal Method

The hydrothermal/solvothermal method usually involves transferring the Mo and S precursor to an autoclave and heating at different temperatures. MoS_2_ nanosheets with different structures can be obtained by adjusting the reactant composition, reaction temperature, time, and pH value [65,66]. This method is easy to control and causes little environmental pollution. In addition, external forces such as microwaves [67,68] and strong magnetic fields [69] are added based on hydrothermal/solvothermal methods with the deepening of research, which accelerates the synthesis of materials and promotes their combination with other components. As shown in Figure 4A, Hao et al. applied the microwave-assisted hydrothermal synthesis method to obtain flower-structured MoS_2_ nanosheets. At the same time, Ni ions were embedded on the MoS_2_ nanosheets to form a unique self-supporting structure. This unique structure effectively inhibited the stacking of MoS_2_ nanosheets [70].

The self-sacrificing template method is also employed to suppress the re-stacking of nanosheets. For example, taking g-C_3_N_4_ as a self-sacrificing template, Huang et al. first obtained a layered MoO_3_/g-C_3_N_4_ precursor with molybdate and g-C_3_N_4_ by hydrothermal synthesis and then processing at 900 °C. The g-C_3_N_4_ template was decomposed, and MoS_2_ nanosheets were finally obtained with a porous ultrathin structure (Figure 4B) [46].

### 3.3. Chemical Vapor Deposition (CVD) Method

Chemical vapor deposition (CVD) is an important synthesis method for 2D materials. Compared with the solvent method and the hydrothermal synthesis method, large-size MoS_2_ nanosheets can be prepared [71]. However, for large-scale synthesis, MoS_2_ often forms a rod shape instead of a sheet shape [71,72,73,74], so high-vacuum reaction conditions are required and precious metal materials such as gold are used as the substrate [75,76,77]. The harsh reaction conditions limit the further application of the CVD method for the synthesis of 2D MoS_2_. To overcome such difficulties, simpler synthesis methods have been further explored. Lee et al. took MoO_3_ and S powder as reactants to directly synthesize MoS_2_ nanosheets on Si/SiO_2_ substrates treated with graphene-like molecules (Figure 5A). Under moderate temperatures, a large area of MoS_2_ layers can be directly obtained on the surface of amorphous SiO_2_ through the reaction of S vapor and MoO_3_. Without using highly crystalline metal substrates or an ultrahigh-vacuum environment, the cost was greatly reduced [71].

Due to the bottom-up synthesis principle of CVD, the continuous monolayer MoS_2_ prepared by CVD is usually atomically neat, and its edges lack saturated atoms as main active sites [78], very unfavorable for the catalytic reaction. To improve this, Tan et al. first prepared porous gold as a template and then synthesized irregularly shaped MoS_2_ nanosheets at the holes of the template (Figure 5B). To further enhance the catalytic activity of MoS_2_ nanosheets, a certain out-of-plane strain on the gold template was introduced so that MoS_2_ was born on the curved inner surface of the template, which gave MoS_2_ a certain degree of strain and improved its hydrogen evolution reaction (HER) catalytic performance [76]. However, this method is still too expensive for mass production; therefore, to reduce the cost, cheaper nanoporous metal templates (such as copper, nickel) could be considered.

## 4. Regulation Strategies toward Oxygen Reduction

### 4.1. 2D MoS_2_ for ORRs

For the ORR, there are currently three mainstream reaction mechanisms: (1) dissociation mechanism, where the O-O bond of oxygen is directly broken when adsorbed on the catalyst surface and O* is reduced into OH* and H_2_O* in the next reaction; (2) association mechanism, where OOH* is first produced during the reaction and then split into O* and OH*; and (3) peroxygen mechanism, in which when the two-electron steps reach OOH* and HOOH*, the latter is decomposed into OH*. The formation of adsorbed oxidizing substances (e.g., O*, OH*, OOH*) caused by the aqueous electrolyte and the reaction itself is considered as a decisive process to determine the performance of the catalyst. All the above three mechanisms indicate that the ORR rate is actually limited by the formation of OOH* and the removal of OH*. The binding energy on the surface of the ORR intermediate should not be too strong or too weak [79]. This phenomenon can be represented by a volcano figure [80,81]. The catalyst located near the apex of the volcano (Figure 6B) theoretically has the best catalytic performance. 

In 2D MoS_2_, when the size of the S-Mo-S three-layer flake is reduced laterally, the coordination number at the edges and corner atoms would become less, which provides rich active sites for electrocatalysis [82,83]. Through theoretical calculations, it can be found that edge-exposed unsaturated Mo atoms are the active sites of ORRs [84]. These unsaturated sites can effectively regulate the adsorption/desorption of reactants and intermediate products on the surface of the catalyst. Although it has been proved that MoS_2_ nanosheets exhibit a certain ORR performance, their reaction path is mostly a two-electron path. At the same time, the catalytic performance of 2D MoS_2_ is lower than that of the commercial Pt/C catalyst and even has a certain gap compared with the popular metal-nitrogen-carbon (M-N-C) catalyst. Two main factors lead to the low catalytic performance of MoS_2_. The first is that pristine 2D MoS_2_ has low conductivity, which is not conducive to the electron transport in ORR processes and thus limits the reaction rate. The second is that, as a 2D material, although pristine 2D MoS_2_ has a large specific surface area, its active sites exist only at the edges and defects of the material, and the huge basal plane is inactive, which greatly hinders the reaction. Thus, how to properly adjust the structure of MoS_2_ to increase the number of active sites and enhance conductivity becomes a key issue in enhancing the catalytic activity of MoS_2_.

### 4.2. Controlling Electronic Structure by Doping

#### 4.2.1. Non-Metallic Element Doping

Studies have shown that incorporating non-metallic elements in MoS_2_ nanosheets can effectively regulate the ORR activity of catalysts due to the changed electronic structures around the doped atoms. It has been demonstrated that the introduction of heteroatoms (O, P, N, etc.) in MoS_2_ can adjust the electronic structure of materials, reduce the bandgap, improve conductivity, and provide catalytically active sites. As presented in Figure 7, Huang et al. used g-C_3_N_4_ as a self-sacrificial template, and through hydrothermal and subsequent heat treatment, MoS_2_ nanosheets with abundant edge defects were obtained, and then oxygen was introduced into MoS_2_ through H_2_O_2_ treatments. The sample with the best catalytic performance had an initial potential of 0.94 V and a half-wave electric potential of 0.80 V in 0.1M KOH. It is interesting to note that the ORR selectivity of O-MoS_2_ changed from a two-electron path to a four-electron path. They believed that this change was due to the highly electronegative O atom polarizing the unsaturated adjacent Mo atom. The Mo atom at the edge generated an additional positive charge and was preferentially adsorbed by oxygen molecules, accelerating the ORR process [51]. Through density functional theory (DFT) calculations, Xie et al. also reported that compared with the original 2H-MoS_2_ nanosheets (1.75 eV), oxygen-bound MoS_2_ nanosheets have a (1.30 eV) narrower bandgap (Figure 7D), indicating that the combination of oxygen and MoS_2_ nanosheets may generate more carriers and increase the intrinsic conductivity of the material [85]. This leads to enhanced ORR and HER activity.

In addition, the incorporation of low-electronegativity atoms such as N and P into MoS_2_ would significantly enhance ORR catalytic activity. However, there are different explanations for performance enhancement. As shown in Figure 8A,B Huang et al. doped P atoms into MoS_2_ nanosheets, which effectively improved the ORR performance of the MoS_2_ nanosheets, and explained the enhancement of ORR performance using boundary molecular orbital theory. Compared to the original MoS_2_ nanosheets, the energy levels of highest occupied molecular orbitals (HOMO) and lowest unoccupied molecular orbitals (LUMO) of P-doped MoS_2_ nanosheets are higher [46]. In semiconductors, increasing the frontier orbital energy is beneficial to electron donation [86], which makes oxygen adsorption and the formation of the intermediate product OH^−^ easier, accelerating the reaction rate.

Zhang et al. calculated the overpotential of N- or P-doped single-layer MoS_2_ in the ORR under acidic conditions and found that replacing S atoms in the MoS_2_ single layer with P or N atoms can introduce high spin density into the basal plane of MoS_2_, thereby improving its ability to activate O_2_ and making the ORR step more inclined to proceed through the more efficient 4e^−^ pathway [87]. It is worth noting that the ORR performance of P-doped MoS_2_ is worse than that of N-doped MoS_2_ due to the excessively high adsorption energy of the intermediate product of P-doped MoS_2_. However, this does not mean that P-doped MoS_2_ has no application prospects. As shown in Figure 8C, through DFT calculations, Liu et al. investigated the ORR mechanism and key active sites of P-doped single-layer MoS_2_ with different structures and the P content in an alkaline environment. It can be seen that double P-doped MoS_2_ with a higher P content had better performance than single P-doped MoS_2_. Its catalytic active site is the S2 atom, which is adjacent to the two P atoms, and the two adjacent P atoms lead to a reduction in the charge of the S2 site and provide a stronger hydrogen bond, boosting the adsorption of H_2_O and OH^−^ groups. Simultaneously, compared with single P-doped MoS_2_, double P-doped MoS_2_ had an appropriate intermediate product adsorption energy (Figure 8D). By DFT calculations, the optimal P doping amount of the catalyst for the ORR was 5.5%, close to that obtained in the experiment (4.7%) [88]. This work provides a reasonable explanation for the contribution of P elements to catalytic activity that has long puzzled researchers.

#### 4.2.2. Metal Element Doping

Metallic elements can also be added to MoS_2_ to strengthen ORR catalytic activity. It has been reported that doping Cu, Co, and other transition metals in the S vacancy of MoS_2_ can effectively enhance ORR catalytic activity. Xiao et al. calculated the theoretical ORR catalytic performance of single-layer MoS_2_ doped with Co/Ni. It is known that doping metal atoms into the basal plane of single-layer MoS_2_ can adjust the adsorption energy of the reaction intermediates on the catalyst surface, thereby increasing catalysis activity. In Co/Ni-doped single-layer MoS_2_, Co/MoS_2_ was similar to the FeN_4_ active site in the M-N-C catalyst, and Ni/MoS_2_ was similar to the CoN_4_ active site. Therefore, the Co/Ni-doped single-layer MoS_2_ had higher ORR catalytic activity than the original MoS_2_ [89]. In addition, Urbanova et al. and He each incorporated metal elements such as Fe, Mn, Ti, and V into MoS_2_ and found a significant improvement in ORR activity (Figure 9A,B) [90,91]. Although much work has been done on the direction of metal doping in MoS_2_, the choice of metal types is a simple try-out method. Using DFT, Wang et al. systematically explored the ORR performance of single-layer MoS_2_ doped with various transition metals. The calculation result showed that transition metal atoms could be embedded in the S vacancy and could significantly change the electronic structure of the material. As shown in Figure 9C,D, they found that Cu-doped MoS_2_ had the best theoretical binding strength for ORR intermediates and therefore had the best ORR catalytic activity, indicating its high potential as an efficient single-atom catalyst for ORRs [92]. This work may serve as a useful guidance for further developing metal-doped MoS_2_-based ORR catalysts.

Since metal-S bonds are regarded as reactive sites in metal-doped transition metal dichalcogenides (TMDs), increasing the density of metal atoms on the surface of MoS_2_ as much as possible and reducing the metal aggregation during heating can be the key to improving the reactivity of the catalyst, but these are difficult to achieve. The metal loading on the surface of most metal-doped MoS_2_ catalysts is very low and difficult to precisely control. Therefore, how to overcome these difficulties is crucial to enhance the ORR performance of metal-doped MoS_2_.

### 4.3. Activating the Basal Plane to Improve the Catalytic Performance of ORRs

To obtain higher ORR performance, many studies have been devoted to optimizing the composition and structure of 2D MoS_2_ through post-processing methods. However, these methods will inevitably destroy the stability of the material and reduce conductivity [93,94]. In fact, the huge impact of the large-area inert basal plane of 2D MoS_2_ on catalytic performance has been neglected for a long time. Thus, how to effectively activate the basal plane of 2D MoS_2_ has become a research hotspot.

#### 4.3.1. Strain Engineering Activates Inert Basal Plane

It has been found that introducing tensile or compressive stress into MoS_2_ can affect the ORR performance. The ultrathin flexible structure of 2D MoS_2_ makes it easier to generate and maintain strain. Through DFT calculations, Zhao et al. observed that tensile strain would shift the center of the p orbital of chalcogen on the surface of TMDs to move to the Fermi level. This shift can enhance the adsorption of materials for ORR intermediates, thereby effectively promoting the ORR process. At the same time, for 2D MoS_2_, the larger biaxial strain would cause a certain degree of lattice mismatch and then inevitably generate new vacancies on the basal plane. The newly generated basal surface vacancies cooperate with strain and further enhance its catalytic performance [95]. These results indicate that strain engineering is an effective way to adjust the ORR activity of 2D TMDs. Recently, a lot of methods have been developed for introducing strain into the material during synthesis, such as using a template with a certain curvature (nanospheres, porous curved metal) or transferring the material to a patterned substrate. For example, Li et al. transferred MoS_2_ to SiO_2_ nanocones by means of capillary pressure and added strain into MoS_2_ (Figure 10) [96]. However, the introduction of strain into 2D MoS_2_ in the experiment to enhance its ORR performance is still not widely used.

#### 4.3.2. Stable Phase Change Activated Inert Basal Plane

Two-dimensional MoS_2_ has different phases, causing various catalytic activities. The 1T phase is a metal phase and has the highest conductivity. However, it is metastable at room temperature and would slowly change into the 2H phase spontaneously. Improving the performance of 2D MoS_2_ by controlling the phase transition has broad research prospects. According to previous reports, the 1T phase can be obtained in a single layer of MoS_2_ through different methods such as alkali metal intercalation-exfoliation and plasma hot-electron injection. In addition, the 1T phase can stably exist by doping Re, Tc, Mn, and other metal elements in nanosheets as electron donors [97,98,99,100]. The high electrical conductivity and catalytic activity make 1T-MoS_2_ a more effective electrocatalyst than 2H-MoS_2_. As exhibited in Figure 11, Sadighi prepared 1T-MoS_2_ nanosheets via the in situ liquid-redox intercalations of sodium ions and solvent exchange exfoliation. The 1T-MoS_2_ nanosheets showed an almost equivalent half-wave potential and enhanced the limit diffusion current as compared to Pt/C [101]. The 1T-phase MoS_2_ nanosheets prepared by the alkali metal intercalation method can spontaneously transform from the 1T phase to the thermodynamically stable 2H phase in the air [59]. This unstable property hinders catalytic stability. At the same time, the conditions of this preparation method are relatively harsh, and the 1T-phase/2H-phase ratio is not high. As a result, it is necessary to develop methods that can easily synthesize 1T-phase MoS_2_ with high stability.

### 4.4. Compounding Other Materials to Enhance Conductivity 

It is well known that the low conductivity and aggregation of nanocrystalline sheets can reduce the catalytic activity of 2D MoS_2_. To avoid these cases, the combination of 2D MoS_2_ with materials possessing high conductivity to form a composite has attracted much attention. Among them, compounding 2D MoS_2_ with carbon materials to form a hybrid structure can effectively boost the conductivity of materials and reduce aggregation [102,103]. As presented in Figure 12A,B, by employing 4-iodophenyl-functionalized monolayer MoS_2_ as a template, Yuan et al. synthesized a series of MoS_2_-coupled sandwich-like conjugated microporous polymers (M-CMPs). After direct pyrolysis, the 2D MoS_2_/N-doped porous carbon (M-CMPs-T) hybrid with a large specific surface area and layered porous structure was finally obtained. Compared with porous nitrogen-doped carbon without MoS_2_, 2D porous carbon hybrids exhibited a higher half-wave potential and limit diffusion current in the ORR [104]. 

Zhao et al. synthesized hybrid nanorods using dopamine and ammonium through self-assembly followed by annealing, while an ultrathin monolayer of MoS_2_ was formed under the constraints of the carbon skeleton (Figure 12C,D). The unique tubular structure, large specific surface area, relatively high N dopant content, and oxygen-bonded MoS_2_ monolayer endowed the MoS_2_/ nitrogen-doped carbon hybrid nanotubes (C HNT) with high electronic conductivity and abundant ORR active sites. The as-prepared MoS_2_/C HNT possessed ORR activity close to commercial Pt/C under alkaline conditions [105]. Zhou et al. used hydrothermal and annealing treatment to attach MoS_2_ nanosheets to graphene oxide (GO), and the interconnected MoS_2_ nanosheets formed a layered structure that completely covered the GO surface. The material revealed excellent ORR performance due to the enhanced conductivity of MoS_2_. At the same time, graphene oxide effectively limited the aggregation of MoS_2_ nanosheets [106] (Figure 12E,F). 

### 4.5. 2D MoS_2_ as a Co-Catalyst

Although 2D MoS_2_ can be directly used as an ORR catalyst, its performance is mostly unsatisfactory due to poor conductivity and rare active sites. Thus, to achieve higher catalytic performance, 2D MoS_2_ can be used as a co-catalyst through having a synergistic effect with other active substances. As shown in Figure 13A–C, Mao et al. used g-C_3_N_4_ as a self-sacrificial template to synthesize ultrathin Ni_3_S_2_/MoS_2_ nanosheets with rich heterogeneous interfaces. The heterostructure formed on the two-phase surface of Ni_3_S_2_ and MoS_2_ generated a large number of unique Mo-Ni-S catalytic active sites. The catalyst revealed higher ORR catalytic activity than the commercial Pt/C [107]. Kwon et al. synthesized an iron phthalocyanine (FePc) and 1T-phase MoS_2_ nanoplate heterostructure catalyst through a one-step hydrothermal reaction. Herein, iron phthalocyanine promoted the transition of nanosheets from the 2H phase to the 1T phase, increasing the electrical conductivity. The unique non-planar geometry of the Fe-N_4_ active site of FePc further enhanced the ORR catalytic activity of FePc-MoS_2_ (Figure 13D,E) [108]. Our group grew 2D MoS_2_ on a N-doped carbon skeleton derived from zeolitic imidazolate frameworks-8 (ZIF-8) and constructed an excellent multifunctional Mo/N/C@MoS_2_ catalyst toward HER, ORR and oxygen evolution reaction (OER). Its excellent catalytic performance originated from the synergistic effect between the abundant main active sites on the edges of MoS_2_, the Mo-N phase coupling center, and the N-induced active sites of nanocarbon frameworks. The porous 3D carbon skeleton structure also accelerated the diffusion rate of products and enhanced electrical conductivity [109].

Based on a strategy derived from ZIF-67, Li et al. synthesized a core-double-shell Co_9_S_8_@Co_9_S_8_@MoS_2_ heterostructure. The Co@Co_9_S_8_ precursor provided sufficient space to limit the growth of MoS_2_, which prevented the aggregation of MoS_2_ and improved the electrical conductivity of the material. The core-double-shell Co_9_S_8_@Co_9_S_8_@MoS_2_ heterostructure exhibited a synergistic interaction between Co_9_S_8_ and MoS_2_ and modified electronic structures. Such characteristics made Co_9_S_8_@Co_9_S_8_@MoS_2_ exhibit excellent electrocatalytic performance, close to commercial Pt/C [111]. As shown in Figure 13F, Vattikuti et al. deposited silver nanocrystals on the surface of 2D MoS_2_ nanosheets by a simple microwave-assisted method. Due to the Ag/MoS_2_ heterostructure and the increased specific surface area, the catalyst presented better ORR activity than pristine MoS_2_ in alkaline solutions [110].

When used as a co-catalyst, 2D MoS_2_ can provide a large-area substrate supporting other active materials (such as precious metals Pt, Ag) and reduce the dosage of active materials. In addition, the heterostructure formed by MoS_2_ and other materials can provide good catalytic activity. Although a variety of MoS_2_-based heterostructure catalysts with excellent ORR catalytic performance have been prepared, the exact reaction that occurs on the heterostructure and the internal mechanism of improving the catalytic performance are still not determined. At the same time, the tunable, controllable, and large-scale synthesis of heterostructures is hard to achieve. 

## 5. Outlook

As a kind of electrocatalytic material with great potential, 2D MoS_2_ has shown great application prospects on the HER, but its development in ORRs is stagnant. Most of the 2D MoS_2_ materials possess low-electron-transport characteristics and a large area of the unused inert basal plane, which limits their applications. This review summarized the research progress in 2D MoS_2_ in ORRs in recent years; focused on the specific works on 2D MoS_2_, from synthesis methods to ORR performance optimization; and identified the key obstacles that 2D MoS_2_ faces in ORRs. To promote the ORR performance of such materials, electron transport and the increase in active sites are basic problems to resolve. Some strategies for resolving these problems, such as increasing exposed edges, heteroatom doping, constructing heterostructures, designing composite materials, and stable phase transitions to enhance electron and product transport, etc., were also discussed in this article. These strategies make it possible to develop high-performance 2D MoS_2_-based ORR catalysts. However, despite numerous encouraging outcomes, there are still many issues to be resolved.

(1) Controllable synthesis. The current bottom-up synthesis method requires relatively harsh conditions, and the top-down synthesis method would cause the aggregation of nanosheets, which lowers the quality of 2D MoS_2_. Striving to achieve large-scale synthesis of high-quality 2D MoS_2_ materials is still the basic requirement.

(2) At present, there is no consensus on the detailed mechanism of 2D MoS_2_ and 2D MoS_2_-based catalysts in the ORR, especially the specific contributions of elements such as O, N, and P. Thus, it is necessary to further analyze the relationship between structure composition and performance to guide future development. 

(3) Effective usage of the large-area basal plane with reactive inertness is necessary. The current work mainly focuses on heteroatom doping and material recombination. However, fine control of heteroatom doping has not been well performed. Further exploring the synergy between MoS_2_ and other ORR active materials, and the internal mechanism of enhanced ORR activity, is the focus for future research.

(4) The intrinsic properties of MoS_2_ can be activated by a stable phase change and strain engineering. Thus, we should find more convenient methods to synthesize long-term stable 1T-phase MoS_2_ nanosheets to improve the adverse effects of spontaneous phase transition on the catalytic performance.

## Figures and Tables

**Figure 1 micromachines-12-00240-f001:**
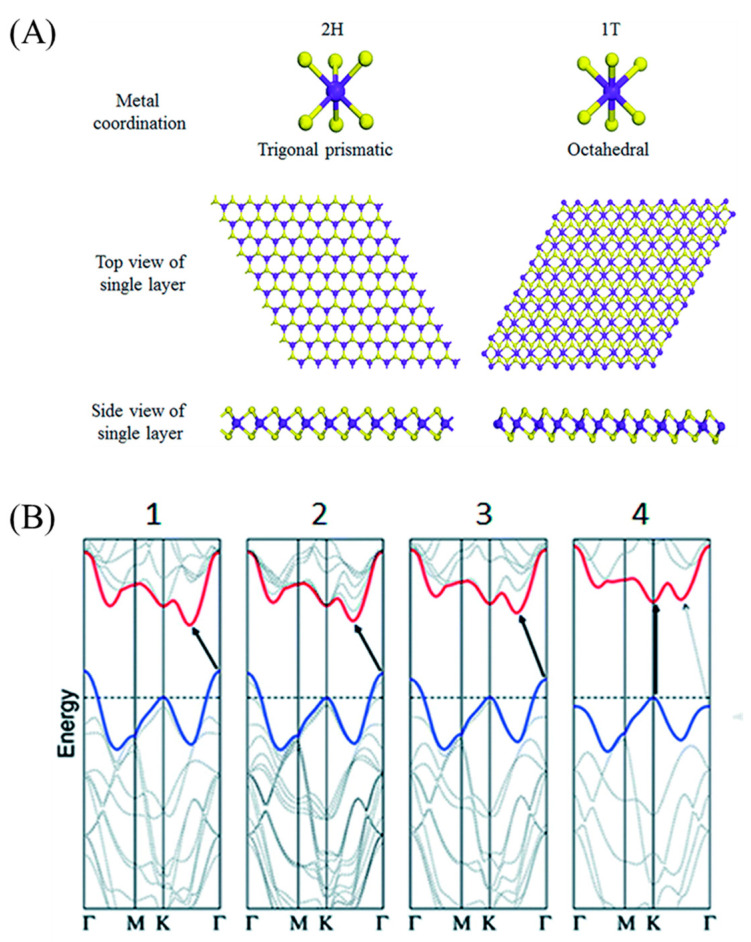
(**A**) Metal coordination, top view, and side view of single-layer 2H-MoS_2_ and 1T-MoS_2_. Reproduced with permission from [44] Copyright © 2021 The Royal Society of Chemistry. (**B**) Energy dispersion (energy versus wavevector k) in (**1**) bulk, (**2**) quadrilayer, (**3**) bilayer, and (**4**) monolayer MoS_2_ from left to right. Reproduced with permission from [48] Copyright © 2021 The American Chemical Society.

**Figure 2 micromachines-12-00240-f002:**
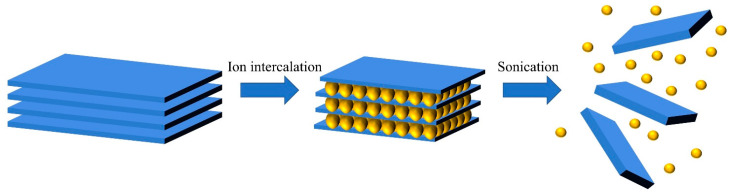
Schematic description of the main liquid exfoliation mechanisms.

**Figure 3 micromachines-12-00240-f003:**
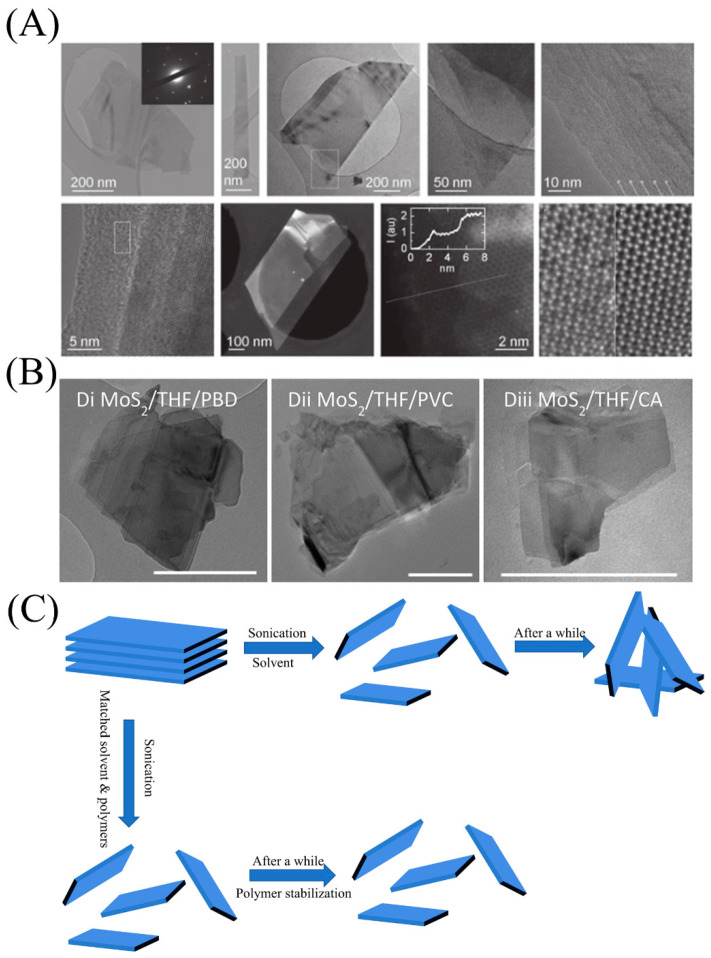
(**A**) Transmission electron microscope (TEM), high-resolution TEM (HRTEM) and high angle annular dark field scanning transmission electron microscope (HAADF STEM) image of MoS_2_ nanosheet exfoliation in sodium cholate solution. Reproduced with permission from [63] Copyright © 2021 WILEY-VCH. (**B**) TEM images of nanosheets dispersed in polymer/solvent solutions. Each image is labeled using the following convention: nanosheet/solvent/polymer. Reproduced with permission from [64] Copyright © 2021 American Chemical Society. (**C**) Schematic diagram of polymer stabilization.

**Figure 4 micromachines-12-00240-f004:**
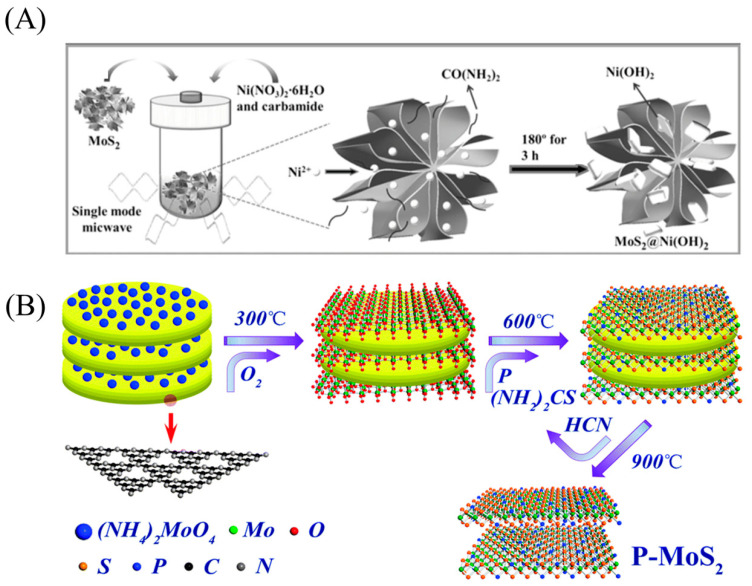
(**A**) Schematic illustration for the formation of self-supporting three-dimensional hierarchical nanostructure MoS_2_@Ni(OH)_2_ nanocomposites. Reproduced with permission from [70] Copyright © 2021 WILEY-VCH. (**B**) Proposed synthetic protocol for ultrathin P-MoS_2_ nanosheets. Reproduced with permission from [46] Copyright © 2021 The Royal Society of Chemistry.

**Figure 5 micromachines-12-00240-f005:**
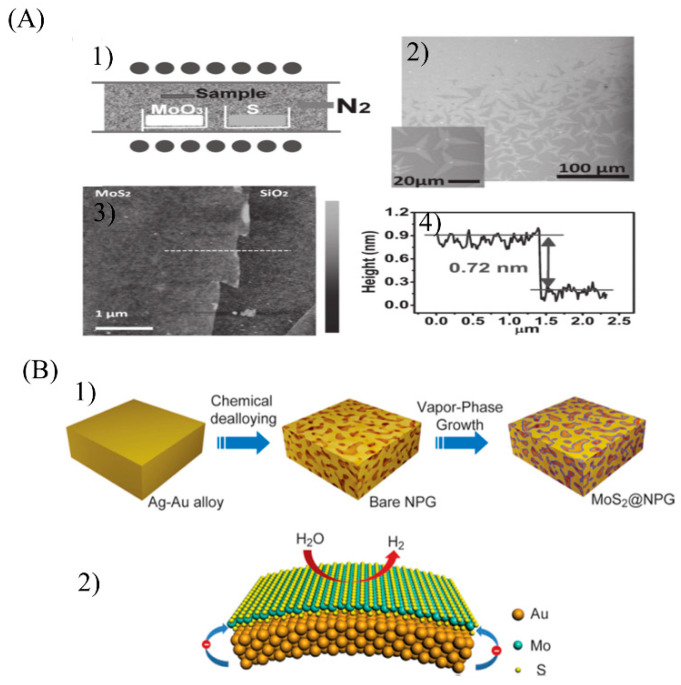
(**A**) (**1**) Schematic illustration of the experimental setup. (**2**) Optical microscopy images of MoS_2_ layers grown on a substrate respectively treated with reduced graphene oxide solution. The inset shows the magnified optical microscopy of the MoS_2_ films, where the seed is observed at the center of each star-shaped sheet. (**3**) Atomic Force Microscope (AFM) image of a monolayer MoS_2_ film on a SiO_2_/Si substrate (pre-treated with reduced graphene oxide (rGO)). (**4**) The thickness of the MoS_2_ layer is 0.72 nm from the AFM cross-sectional profile along the line indicated. Reproduced with permission from [71] Copyright © 2021 WILEY-VCH. (**B**) Monolayer MoS_2_@nanoporous gold (NPG) toward catalytic hydrogen evolution reaction (HER). (**1**) Schematic diagram of the fabrication process of monolayer MoS_2_@nanoporous gold (NPG) hybrid materials by a nanoporous metal-based CVD approach. (**2**) Schematic HER catalyzed by the monolayer MoS_2_@NPG hybrid material. Reproduced with permission from [76] Copyright © 2021 WILEY-VCH.

**Figure 6 micromachines-12-00240-f006:**
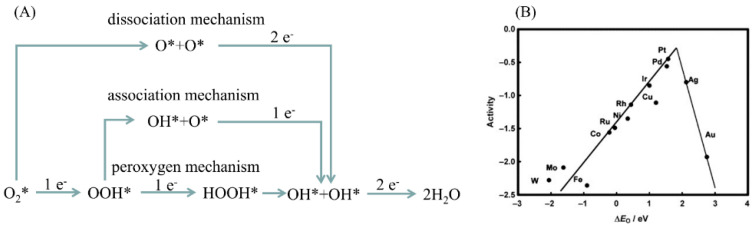
(**A**) Proposed oxygen reduction reaction (ORR) mechanisms. Reproduced with permission from [80] Copyright © 2021 WILEY-VCH. (**B**) Volcano-type relationship for the ORR activity versus the oxygen-binding energy. Reproduced with permission from [81] Copyright © 2021 The American Chemical Society.

**Figure 7 micromachines-12-00240-f007:**
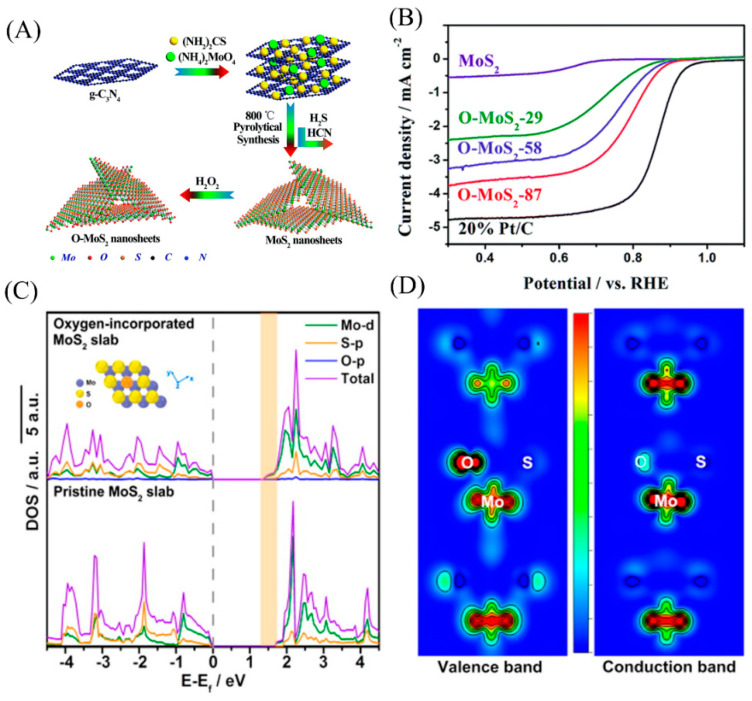
(**A**) Proposed synthetic protocol for incorporation of oxygen into ultrathin MoS_2_ nanosheets. (**B**) ORR performance test for O-MoS_2_ and pristine MoS_2_ nanosheets as well as 20% Pt/C in oxygen-saturated 0.1 M KOH. Reproduced with permission from [51] Copyright © 2021 The Royal Society of Chemistry. (**C**) Calculated density of states (DOS) of the oxygen-incorporated MoS_2_ slab (top) and the pristine 2H-MoS_2_ slab (bottom). The orange shading clearly indicates the decrease in bandgap after oxygen incorporation. (**D**) The charge density distributions of the valence band (left) and conduction band (right) near the oxygen atom in oxygen-incorporated MoS_2_ ultrathin nanosheets. Reproduced with permission from [85] Copyright © 2021 The American Chemical Society.

**Figure 8 micromachines-12-00240-f008:**
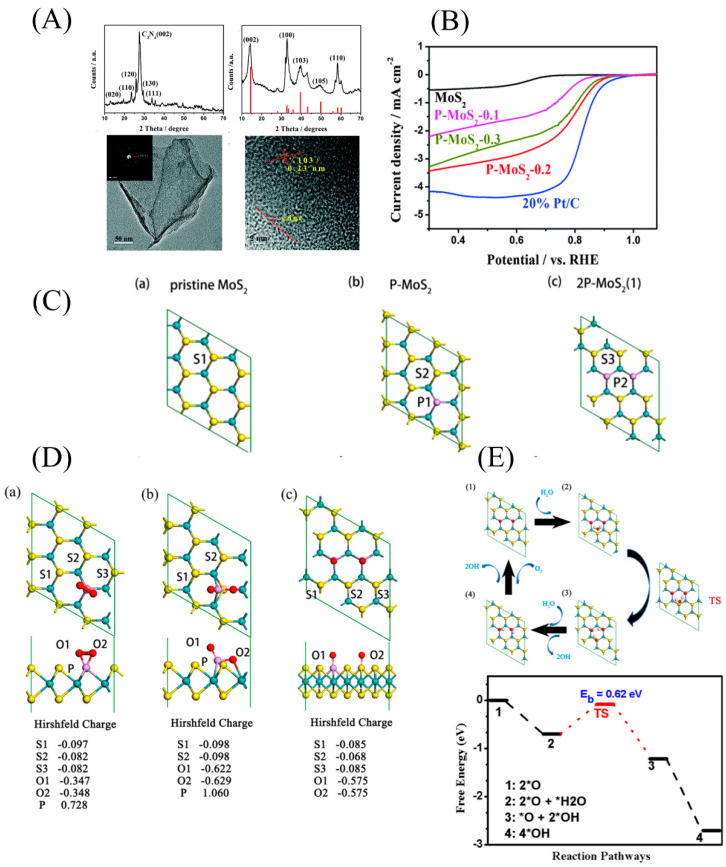
(**A**) X-ray diffraction (XRD) patterns of the MoO_3_/g-C_3_N_4_ intermediate and ultrathin P-MoS_2_ nanosheets and TEM and high resolution-TEM images of ultrathin P-MoS_2_ nanosheets. (**B**) ORR electrocatalysis for P-MoS_2_-0.1, P-MoS_2_-0.2, P-MoS_2_-0.3, MoS_2_, and 20% Pt/C catalysts. Reproduced with permission from [46] Copyright © 2021 The Royal Society of Chemistry. (**C**) Top view of the optimized structure of (**a**) pristine MoS_2_ nanosheets, (**b**) single P-doped MoS_2_ nanosheets, and (**c**) double P-doped MoS_2_ nanosheets. (**D**) Top and side views of the optimized adsorption configurations of (**a**) O_2_-adsorbed P-MoS_2_, (**b**) 2O-adsorbed P-MoS_2_, and (**c**) 2O-adsorbed 2P-MoS_2_. The Hirshfeld charge values of the corresponding atoms are also given. (**E**) Optimized structure configuration of reactant, intermediates, and product (**1**–**4**) and the corresponding reaction pathways of the ORR on a 2P-MoS_2_ sheet in an alkaline environment. Reproduced with permission from [88] Copyright © 2021 American Association for the Advancement of Science.

**Figure 9 micromachines-12-00240-f009:**
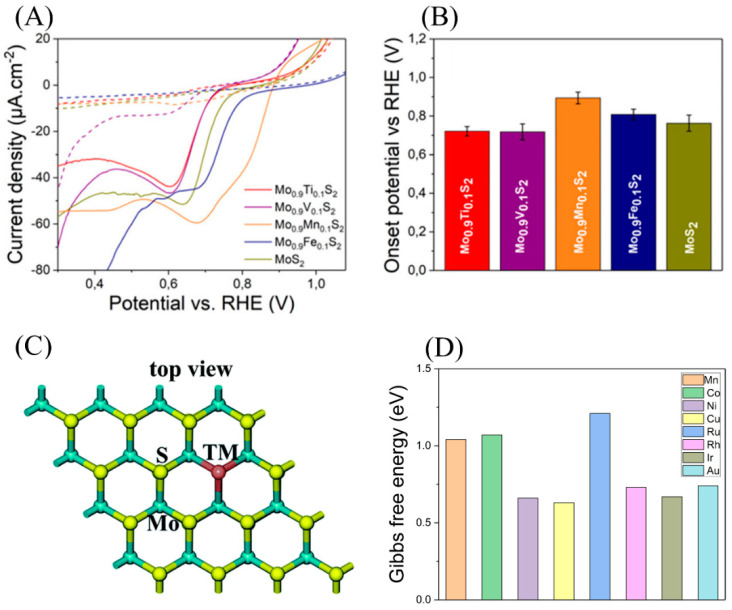
(**A**) Typical linear sweep voltammetry (LSV) curves toward ORRs with the corresponding values of onset potentials (**B**). Reproduced with permission from [90] Copyright © 2021 The American Chemical Society. (**C**) Schematic of a transition atom that embeds into the S vacancy of the MoS_2_ monolayer. Reproduced with permission from [92] Copyright © 2021 The Royal Society of Chemistry. (**D**) Schematic of Gibbs free energy diagrams of the ORR on various transition metal embedded MoS_2_ monolayers via the OOH association mechanism in an acidic medium. Based on literature data from [92].

**Figure 10 micromachines-12-00240-f010:**
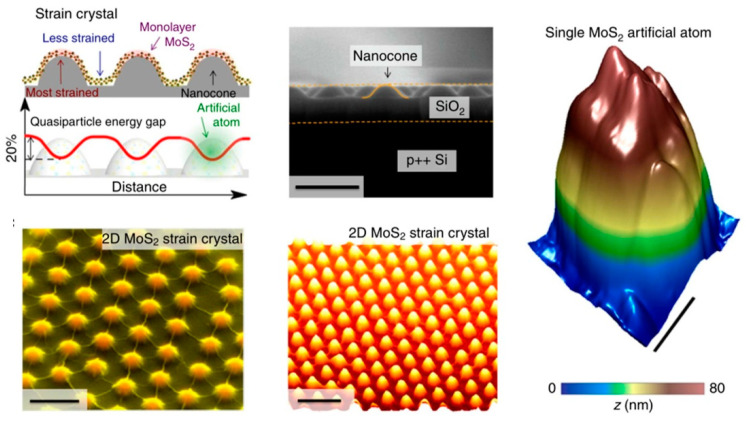
Assembly of an artificial-atom crystal via spatially patterned biaxial strain within a monolayer of MoS_2_. Reproduced with permission from [96] Copyright © 2021 Nature Publishing Group.

**Figure 11 micromachines-12-00240-f011:**
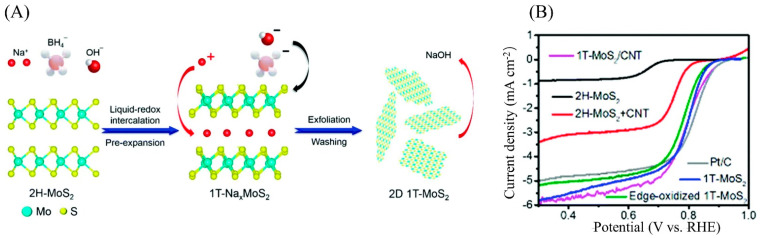
(**A**) Illustration of the synthetic procedure. (**B**) Electrocatalytic activities of 1T-MoS_2_, 1T-Na*_x_*MoS_2_, 1T-MoS_2_/carbon nanotube (CNT), 2H-MoS_2_, 2H-MoS_2_/CNT, and Pt/C in O_2_-saturated 0.1 M KOH. Reproduced with permission from [101]. Copyright © 2021 The Royal Society of Chemistry.

**Figure 12 micromachines-12-00240-f012:**
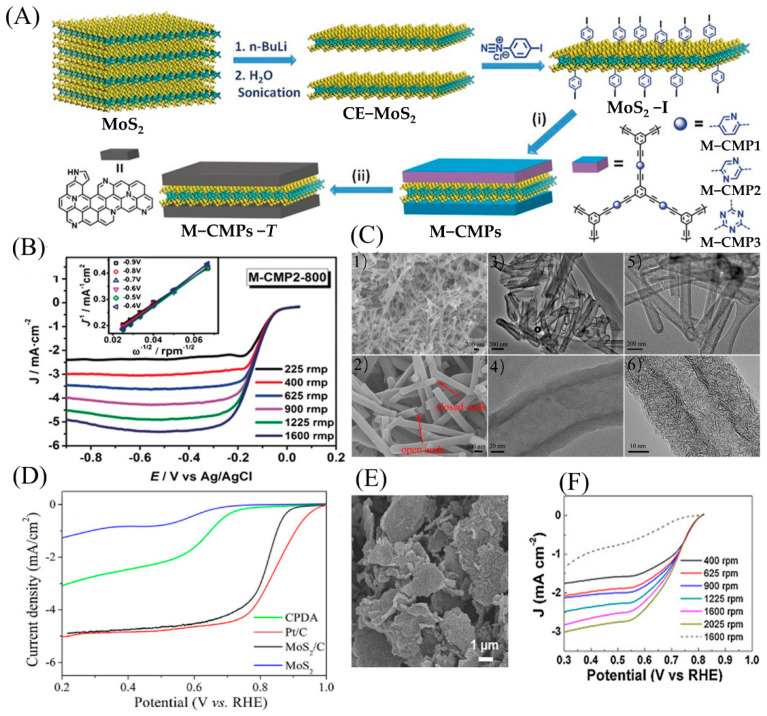
(**A**) Idealized formula scheme depicting the chemical exfoliation of bulk MoS_2_ and subsequent functionalization with 4-iodophenyl substituents under the formation of MoS_2_-I as well as the preparation of MoS_2_-templated conjugated microporous polymers and the corresponding MoS_2_/nitrogen-doped porous carbon hybrids. (**B**) Electrochemical evaluation of catalysts in alkaline media of the MoS_2_-based porous carbon nanosheets. Reproduced with permission from [104] Copyright © 2021 WILEY-VCH. (**C**) (**1**), (**2**) Scanning electron microscope (SEM), (**3**), (**4**) TEM images of (NH_4_)_2_MoS_4_/PDA nanotubes; (**5**) TEM image and (**6**) HRTEM image of MoS_2_/ nitrogen-doped carbon hybrid nanotubes (C HNT). (**D**) Electrochemical evaluation of catalysts in alkaline media of the Pt/C, CPDA, bulk MoS_2_, and MoS_2_/C HNT. Reproduced with permission from [105] Copyright © 2021 American Chemical Society. (**E**) Field emission-SEM images of hierarchical MoS_2_− reduced graphene oxide (rGO) nanosheets. (**F**) LSV curves of MoS_2_–graphene oxide (GO) (solid curves) and MoS_2_ microparticles (dashed line) regarding the ORR in O_2_-saturated 0.1 M KOH at various rotating speeds. Reproduced with permission from [106] Copyright © 2021 Elsevier B.V.

**Figure 13 micromachines-12-00240-f013:**
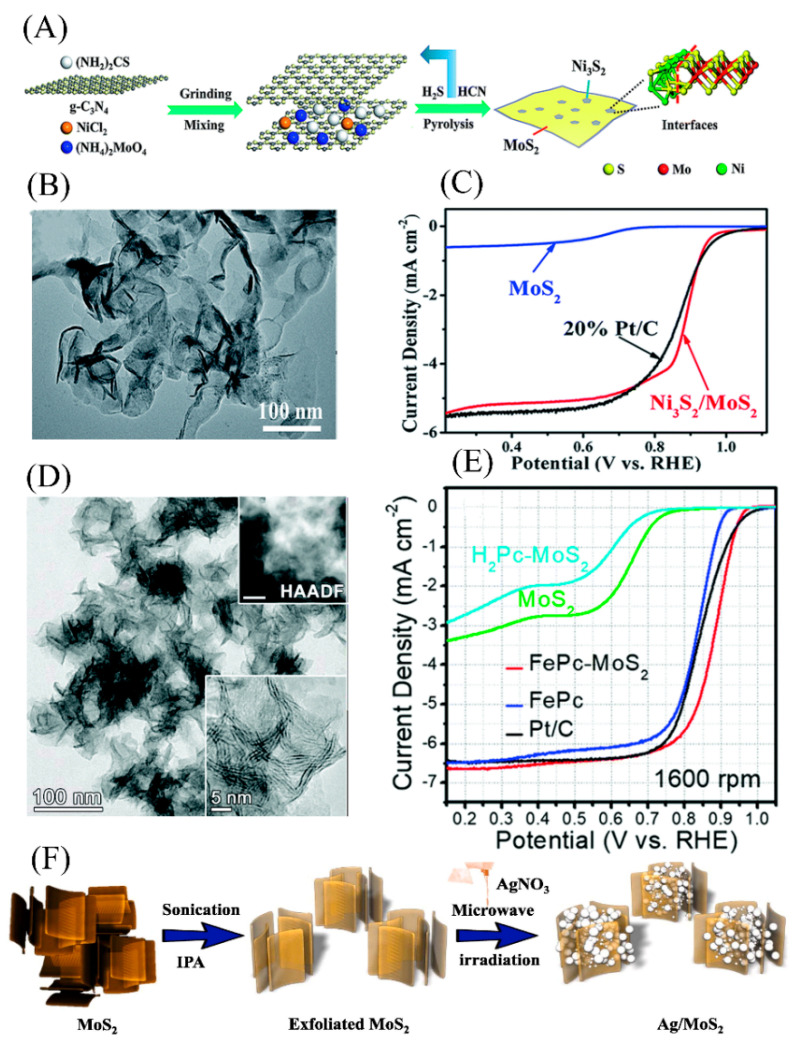
(**A**) Proposed synthetic protocol for ultrathin Ni_3_S_2_/MoS_2_ nanosheets. (**B**) TEM image of Ni_3_S_2_/MoS_2_-0.2 nanosheets. (**C**) Electrochemical evaluation of catalysts in alkaline media. Reproduced with permission from [107] Copyright © 2021 The Royal Society of Chemistry. (**D**) HRTEM and HAAD-STEM images of the FePc-MoS_2_ hybrid complex. (**E**) Electrochemical evaluation of catalysts of FePc-MoS_2_ in alkaline media. Reproduced with permission from [108] Copyright © 2021 The Royal Society of Chemistry. (**F**) Representation of synthetic procedure of a Ag/MoS_2_ nanohybrid. Reproduced with permission from [110] Copyright © 2021 Elsevier B.V.

## Data Availability

No new data were created or analyzed in this study. Data sharing is not applicable to this article.

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
