# Peer review of "Two-Dimensional MoS2: Structural Properties, Synthesis Methods, and Regulation Strategies toward Oxygen Reduction"

_micromachines, 2021, doi:10.3390/mi12030240_

Round 1

Reviewer 1 Report

In the current review, Xu et al. have reviewed the MoS2 electrocatalyst for ORR applications and discussed its structure, synthesis, and modification strategies at length. As the authors mentioned, there are only a very few reviews in the case of ORR, especially with  MoS2 electrocatalyst. In this respect the review is interesting and well organized; however, I have a few concerns/suggestions that need to be addressed before the consideration of the manuscript for publication.

  1. The manuscript requires thorough English/ grammatical corrections. Certain sentences are very hard to follow and the readers have to just assume what the authors are trying to convey. For instance, “ During the process from bulk to single-layer TMDs”: do the authors mean “During the conversion process of TMDs from bulk to single-layer”? (It is unclear what process the authors are referring to). Such discrepancies can be found throughout the manuscript. Also, there are several redundant sentences that can be avoided to improve the quality of the manuscript.
  2. The authors have described the role of MoS2 in ORR pages 3 and 4 (lines 119- 142). As the whole review is centered on the electrocatalytic properties of MoS2, it would be appropriate to describe why MoS2 is important (what distinguishes it from other TMDs). The reviewer believes such information is crucial for this kind of specific review articles.
  3. Line 153: are the authors sure it is “n-but lithium”, all the abbreviations are suggested to be rechekced.
  4. The work comprises some published figures with little to no new schematics or figures, to represent the information more simply and effectively. It would be informative if the authors provide more illustrative figures from their perspective.
  5. Overall, the clarity of all the images can be improved.
  6. It is well accepted in the catalysis community and the authors also pointed out the importance of the “volcano figure” (Page 8). It is suggested to provide a relevant volcano plot in the manuscript.
  7. The electron transfer process, specifically the number of electrons transferred is critical in the ORR system. The authors vaguely introduced “ two-electron paths” (Page 9- line 279). It would be helpful for the readers if the number of electrons is represented in scheme 1 or some other representation.
  8. In the introduction, the authors mention “These special properties of TMDs nanosheets provide a valuable source of inspiration for basic research in many fields including catalysis[28,29], transistor[30], energy storage[31,32]and sensor[33,34].” Some related references are suggested to be included here, Catalysis: (ACS applied materials & interfaces 10 (33), 27771-27779; Sustainable Energy & Fuels 2 (1), 96-102), energy storage (ACS Applied Materials & Interfaces 12 (24), 27112-27121; J. Am. Chem. Soc. 2017, 139, 1, 171–178; The Journal of Physical Chemistry C 121 (23), 12718-12725) and sensors (Biosensors and Bioelectronics 172, 112724).

Reviewer 2 Report

This paper is interesting. The author reviewed the state of the art of Two-dimensional MoS2 in structural properties, synthesis meth-2 ods and regulation strategies for ORR. Before publication, those comments should be addressed.

  1. please double check all citation in this paper. There should be a space after the citation. For example, in line 179, the citation [46, 54] is OK. But it’s not good for [53]. Because there should be a space between [53] and ‘and’. Please fix all problems in this paper.
  2. Generally the paper is poorly written and very hard to follow. Please have this paper English proofreading.
  3. more references are needed. Check for example the following references:

Mao et al., The rise of two-dimensional MoS 2 for catalysis

Rafael A Vilá et al., In situ crystallization kinetics of two-dimensional MoS2

Wei et al. Ultra-high-aspect-ratio vertically aligned 2D MoS2-1D TiO2 nanobelt heterostructured forests for enhanced photoelectrochemical performance

Xiao Li et al., Two-dimensional MoS2: Properties, preparation, and applications

Round 2

Reviewer 1 Report

The authors have now carefully addressed most of the reviewer's comments. Hence, I recommend the acceptance of this manuscript in the current form. 

Reviewer 2 Report

The authors have addressed all the comments. However, the language of this paper is still poor. It requires moderate English polishing.